# Does Self-Directed Learning with Simulation Improve Critical Thinking and Motivation of Nursing Students? A Pre-Post Intervention Study with the MAES© Methodology

**DOI:** 10.3390/healthcare10050927

**Published:** 2022-05-18

**Authors:** Vanessa Arizo-Luque, Lucía Ramirez-Baena, María José Pujalte-Jesús, María Ángeles Rodríguez-Herrera, Ainhoa Lozano-Molina, Oscar Arrogante, José Luis Díaz-Agea

**Affiliations:** 1Nursing Department, Catholic University of Murcia (UCAM), 30107 Guadalupe de Maciascoque, Spain; varizo@ucam.edu (V.A.-L.); jluis@ucam.edu (J.L.D.-A.); 2Red Cross University Centre for Nursing, University of Seville, 41009 Sevilla, Spain; 3Department of Health Sciences, Faculty of Health Sciences, Universidad Europea de Valencia, 46010 Valencia, Spain; mariaangeles.rodriguez@universidadeuropea.es; 4University School of Nursing of Ávila, Department of Nursing of the University of Salamanca, 05003 Ávila, Spain; ainhoa.lozano.molina@usal.es; 5Red Cross University College of Nursing, Spanish Red Cross, Autonomous University of Madrid, 28003 Madrid, Spain; oscar.arrogante@cruzroja.es

**Keywords:** motivation, critical thinking, simulation-based education, MAES methodology, nursing students

## Abstract

Motivation and critical thinking are fundamental for the development of adequate learning. The purpose of the present study was to assess the motivation for learning and critical thinking among nursing students before and after self-directed simulation-based training using the MAES© methodology. A cross-sectional and descriptive quantitative study was conducted with a sample of third-year nursing students. The instruments utilized were the Spanish-adapted version of the Motivated Strategies for Learning Questionnaire (MSLQ-44), and the Critical Thinking for Nursing Professionals Questionnaire (CuPCPE). The students improved their levels both of motivation components, (such as self-efficacy, strategy use, self-regulation) and critical thinking components (such as personal characteristics, intellectual and cognitive abilities, interpersonal abilities and self-management, and technical abilities). These improvements could be a result of the intrinsic characteristics of the MAES© methodology (as a team-based, self-directed, collaborative and peer-to-peer learning method).

## 1. Introduction

MAES© is the Spanish acronym of the Self-Learning Methodology in Simulated Environments. This learning method using simulation was developed by a team of educators at the Catholic University of Murcia in 2013 [1,2]. The Self-Learning Methodology in Simulated Environments (MAES©) is an active teaching–learning method involving high-fidelity clinical simulation. Work is conducted with small groups of participants guided by a facilitator (a maximum of 12 to 18 students) in a simulated, psychologically-safe environment [3], and with specific guidelines in which the group and work teams that comprise it have autonomy, identity, ability to make decisions, and high commitment to collaborative learning [4,5].

The seven basic principles of good teaching practices are the result of studies centered on teachers and students, underlining the need to stimulate cooperation between students, promote active learning, and project expectations onto students, considering different talents and manners of learning [6]. Therefore, the profile of the MAES© facilitator [7] is that of an educator who motivates the search for knowledge and the autonomy of the group of students. The MAES© methodology facilitator learns from the students and values the bi-directional character [7,8] of learning obtained from their interaction in the management of the clinical simulation. The model of learning which fits the MAES© facilitator is based on the constructivist framework [8], and is in agreement with a profile that is different from a provider of information [9,10,11], framed within an area closely related to motivational aspects through experiential learning [12].

This methodology is composed of six stages, and allows the acquisition of technical and non-technical skills in a manner that is experiential and reflective [8]. The first stage of the MAES© methodology is the selection of work groups and the establishment of a group identity through structured and guided group dynamics. The second is the voluntary selection of the study subject, starting with cases extracted from reality or fiction. The third is the establishment of baseline competences and the programming of competences sought, through a brainstorming session as a group. The fourth stage is students’ design of a clinical simulation session, to explore the competences sought. The fifth refers to the execution of the clinical experience simulation. Finally, the sixth stage of MAES© is debriefing after the simulation experience, along with exposition of the competences acquired (the expository phase of the Debriefing).

Prior studies on learning with MAES© have described certain advantages over other styles of learning that include simulation [13], especially in the acquisition of technical skills, and overall with non-technical skills such as communication. Also, better scores have been described for the learning of clinical skills related to surgical safety in perioperative nursing [14] compared with traditional learning through practical seminars. Likewise, the value of MAES© for the training of recently-hired nurses in an emergency department in the United Kingdom has also been described [15].

In various prior qualitative studies of this method [4,5,7,8,13] the importance of motivation has been underlined, as well as how information is integrated through cognitive and affective-motivational processes, improving the method’s influence in educational contexts. For the intrinsic motivation of the students, the MAES© methodology works with subjective factors such as curiosity, perceived incongruence or control, which together with immediate feedback, improves the concept of flow and increases mastery of the activity.

Other aspects to have emerged from prior research on MAES© include the subjective perception of acquisition criteria when participants make non-directed decisions, along with centering in scientific evidence the discussion on the behaviors in the simulation [14]. These direct our attention to motivation and critical thinking as fundamental dimensions of the MAES© methodology, which have an impact on the improvement of emotional intelligence, as the student must perceive, understand, put into practice, and self-regulate to acquire the competences sought [16].

Critical thinking, when focusing on the resolution of problems, allows difficulties to create a triple response in students, that is, becoming aware of what is learned, reflection, and learning in a context that is as close as possible to real life, so that the knowledge and skills developed have a greater probability of being used, and so that this arrangement or transfer of knowledge becomes useful [17]. Furthermore, a learning community based on dialogic education is created [18], in which the student is placed at the center, starting with the student’s own baselines, and working on competences from the objectives they themselves have defined.

Given the above, the following research question guided this study: How does MAES© training influence nursing students in areas such as motivation for learning and critical thinking? To answer this question, the following objectives were set:To evaluate the learning motivation strategies of nursing students before and after the self-directed simulation training program (MAES©).To analyze the critical thinking of nursing students before and after the self-directed simulation training program (MAES©).

## 2. Materials and Methods

### 2.1. Design

A multi-center, pre-post cross-sectional and descriptive quantitative study was conducted without a control group.

### 2.2. Participants

A universal sample was chosen comprised of third-year nursing students from two Spanish universities, who were about to start simulation sessions with the MAES© methodology during the 2019–2020 academic year. The simulation groups had already been defined according to the group-making guidelines at their respective centers, so randomized group distribution was not performed. The universities were the Red Cross University College of Nursing at the University of Seville, and the European University of Valencia. From a total of 150 students enrolled in the third year of the Nursing Degree at the two centers, a final sample of 77 participants was obtained, at a response rate of 51.33%. The loss of participants can be explained by the peculiarities of the COVID-19 pandemic (decreased in-person attendance), together with the months that passed between the pre- and post-tests.

The inclusion criteria were: being in the third year at the centers included in the study; enrolment in the courses in which the MAES© methodology was utilized during the 2019–2020 academic year; not being enrolled in student exchange programs (such as ERASMUS), which could interfere with the collection of data.

### 2.3. Study Variables

The independent variable evaluated was the training with the MAES© methodology, and the dependent variables were motivation and critical thinking, which were measured with validated instruments. Also, sociodemographic variables were collected, such as: age, gender, clinical experience, previous education, and previous learning with simulation.

### 2.4. Instruments

The questionnaire for measuring motivation was the Spanish version [19] of the Motivated Strategies for Learning Questionnaire (MSLQ-44) [20], and for critical thinking was the N-CT-4 Practice (Nursing Critical Thinking in Practice Questionnaire), the Spanish version of which is the “Critical Thinking Questionnaire for Nursing Professionals” (CuPCPE) [21].

The Motivated Strategies for Learning Questionnaire (MSLQ) [19], named in Spanish [20] “Cuestionario de Estrategias de Aprendizaje y Motivación” (CEAM II), is based on the self-regulated learning methodology [22,23]. Thus, the MSLQ is based on the self-regulated learning theoretical framework from Paul Pintrich, itself based on the socio-cognitive theory of learning motivation and self-regulation, which assumes that these processes are not characteristics exclusive to the student but are also dependent on numerous factors involved in the learning environment. There is a long version of the questionnaire, with 81 items, as well as a short version with 44 items (MSLQ-44), which was utilized in the present study. This instrument, divided into two scales, allows information to be obtained relating to motivation and cognition, each divided into five subscales (self-efficacy, intrinsic value, test anxiety, cognitive strategies, and self-regulation or metacognition) throughout the 44 items, scored with a Likert scale from 1 to 7, with 1 being “not at all true for me” and 7 “very true for me”. The internal consistency of all the subscales from the original version of the MSLQ-44 [24] was satisfactory, with values oscillating between 0.74 and 0.89. More specifically, the Cronbach alpha values for each of the subscales were: self-efficacy (0.89), intrinsic value (0.87), test anxiety (0.75), use of learning strategies (0.83) and self-regulation (0.74).

The Critical Thinking for Nursing Professionals Questionnaire (CuPCPE) [21] is based on the Alfaro-LeFevre 4-Circle CT Model, which defines critical thinking using 109 items with a Likert scale from 1 to 4, where 1 is “never or almost never” and 4 is “always or almost always”, applied to four dimensions: (a) personal, (b) intellectual and cognitive, (c) interpersonal and self-management, and (d) technical. It has an internal consistency with Cronbach’s alpha of 0.96. For each dimension, the values oscillated between 0.78 and 0.94. The time needed for its completion was about 25 min. The total scores oscillated between 109 and 436. For each dimension, there is also a specific score. A greater score indicates that the student possesses high levels of critical thinking.

### 2.5. Procedure

Before the study, the facilitators in charge of the simulation took part in a training session on the MAES methodology at the Catholic University of Murcia (Spain). They had the opportunity to use this method with nursing degree students, and took the MAES© methodology facilitator course (this is a standardized course available at https://moocucam.appspot.com/maes/preview) (accessed on 1 May 2021). They were tutored the entire time and acquired the necessary competences for conducting the simulations. Afterwards, in the study design phase, a common strategy was created so that both centers could follow the same stages of the method in a rigorous manner. Next, a feedback session about the development of the sessions took place with the project director and the team of researchers.

Pre-test: The questionnaires were provided to all students enrolled before the first MAES© session, through an ad placed on the virtual campus website. They were informed that it was important for them to complete it before the start of the sessions, and they were encouraged to read and sign the consent form. Next, various sessions were conducted to cover the MAES stages. In the first session, a pre-briefing was conducted, in which all the pre-session elements were utilized [25], a safe environment was established, the work teams were defined, the fictional contract was described, and the learning objectives were decided according to the students’ preferences. Also, a commitment was made to resolve the deficits in knowledge based on the baseline competences of the group. Lastly, the bases for the designs of the scenarios were defined, and the participants were trained with examples.

In two sessions after the pre-briefing, the students worked on the simulation scenarios designed by the different teams, and they were described with scientific evidence. Each scenario was followed by a debriefing session, and a presentation of the competences was given by the teams, under the supervision and guidance of a facilitator. Each student participated in three sessions lasting 4 h each. Manikins and standardized patients were utilized depending on the scenarios chosen by the students. In the case of scenarios that included procedures such as cardiopulmonary resuscitation, a manikin was used. However, in scenarios in which non-technical skills were worked on, standardized patients were utilized (the actors were the students who had designed the active case).

The questionnaires were again provided after the last MAES experience ended. The students were given a week to complete and submit both questionnaires through an online questionnaire platform.

### 2.6. Data Analysis

Study data were analyzed using IBM SPSS version 20.0. Descriptive statistics including the frequencies, percentages, means, and standard deviations for the overall scales were calculated. In addition, paired tests for the pre- and post-simulation *t*-test scores were also calculated. The null hypothesis proposed is the inexistence of a change in motivation and critical thinking in students after the application of MAES; the alternative hypothesis is that the MAES methodology increases both qualities in students. For all the variables, including the totals calculated, normality was analyzed using the Kolmogorov-Smirnov test. The percentage of change pre- and post-simulation was calculated, considering the total possible score for each section of the scale.

### 2.7. Ethical Considerations

Institutional review board approval was obtained prior to study implementation (UCAM ethics committee—Ref number: CE012107). The simulation was conducted as a required class activity, but the pre- and post-surveys were completed by the participants on a voluntary basis. The participants gave their informed consent if interested in participating, and they were informed that their information would be kept confidential. The original author’s permission was obtained for the use of the CuPCPE and MSLQ44 scales. The study complies with Spanish laws, established in Royal Decree 1791/2010, from December 30th, which approves the Statute of the University Student.

## 3. Results

Table 1 shows the general characteristics of the 77 study participants, of which 56 completed both questionnaires, 5 participated only in the CuPCPE questionnaire, and 16 completed only the MSLQ-44 questionnaire.

Most of the students were women (87%), with a mean age of 22 (SD 2.2), without prior experience with simulation (82%), clinical experience (75%) or prior university education (47%). Twenty-one students completed only one of the two requested questionnaires. These students did not differ in age, gender or previous clinical experience with those who filled out both questionnaires; however, we can observe that most of the students with prior experience in simulation completed only one questionnaire (Fisher’s test—*p* < 0.001), and that these had, more often than not, previous education (Fisher’s test—*p* < 0.001).

As described in Table 2, for the 61 students who answered the CuPCPE questionnaire, the total score was calculated per dimension before and after the simulation, through the sum of the scores from the questions included in each dimension (intellectual and cognitive, personal, interpersonal, and self-management, and technical), as well as the total score of the questionnaire pre- and post-simulation. All the variables, including totals, complied with the normality criteria. When comparing the pre- and post-intervention scores, the questionnaire indicated a significant increase in the total (4%), and all the dimensions, especially the interpersonal and self-management (6.4%) and the intellectual and cognitive (5%). When examining the test scores of the students who answered only the CuPCPE questionnaire and those who answered both, no significant differences were observed in the totals, neither according to the sections, nor in the total result, nor between pre- and post-simulation scores.

For the 71 students who completed the MSLQ-44 questionnaire (Table 3), the total score was calculated before and after the simulation, by adding the scores from the questions included in the questionnaire. The pre- and post-simulation scores were also calculated for each scale (self-efficacy, intrinsic value, test anxiety, use of learning strategies, self-regulation) and for each section (Motivation and Learning Strategies). All variables, including the totals, complied with the normality criteria. When comparing the pre- and post-intervention scores, a significant increase was observed in the motivation of the students, with a mean of 5.5 points (*p* = 0.003). Within the Motivation section, an increase was observed only in the self-efficacy scale (2.1%), as opposed to in the Learning Strategies section (an increase of 2.5%), where improvement was observed in all the scales after the simulation, especially in the use of learning strategies. When examining the test scores of the students who answered only the MSLQ-44 questionnaire and those who answered both, no significant differences were observed in the totals, neither according to the sections, nor in the total result, nor between the pre- and post-simulation scores.

## 4. Discussion

Previous studies about learning with the MAES© methodology have shown its effectiveness in improving both technical and non-technical skills, highlighting the relevance of nursing students’ motivation [4,5,7,8,13,15] and critical thinking [14]. However, these dimensions have not to date been analyzed through quantitative methods. After conducting quantitative analysis in this study, our results show the positive effects of the MAES© methodology on developing both motivation and critical thinking in undergraduate nursing students. Therefore, the results provide an answer to our research question, as the training with the MAES© methodology had a positive influence on motivation for learning, as well as on the critical thinking of the nursing students. For motivation, our results showed that the nursing students improved their total scores in this dependent variable. Likewise, they improved their levels of the three components of motivation: the self-efficacy component, and those of strategy use and self-regulation, which are part of self-regulated learning. The results obtained in our study are congruent with previous studies that also applied the MSLQ-44 [19,24] to assess motivation, and revealed significant improvement in motivation due to exposure to high-fidelity simulation [26,27].

Focusing on the components of motivation that the nursing students improved, we find that they achieved significant levels of self-efficacy. This competency consists of a future-oriented, optimistic belief that increases motivation, equating to improved performance [28]. Self-efficacy is considered a healthcare professional skill for successfully managing complex and stressful situations [29]. In this sense, there is ample evidence in the literature that suggests simulation is effective at increasing this competency [30]. Particularly, single-group pre-test and post-test design studies have reported increases in self-efficacy after simulation sessions using standardized patients [31,32,33].

Likewise, the MAES© methodology also had a positive influence on the two components of self-regulated learning, given that our nursing students improved their levels of strategy use and self-regulation after the MAES© sessions. This implies that they developed their cognitive and self-regulation strategies, which allowed them to collect, analyze, process, and apply all the information required to achieve their academic learning. This result is in agreement with previous studies that applied effective instructional interventions, including a clinical simulation methodology, which resulted in the improvement of the academic motivation of nursing students [34,35].

We must underline that motivation is considered a key aspect of the learning and educational performance of nursing students [36]. In this sense, motivated students tend to be more engaged with learning activities, choose appropriate learning and studying styles, ask for help if needed, pay more attention to curriculum activities, and consequently are more successful in educational environments [34,35]. According to the recent review conducted by Saeedi et al. [36], the clinical simulation methodology including high-quality simulation and using a standard patient is considered an effective instructional intervention that improves academic motivation in nursing students [26,27,37,38,39]. The MAES© methodology meets all the requirements defined by the INACSL Standards of Best Practice: Simulation^SM^ [40] for conducting high-fidelity simulation sessions, and standardized patients participated in the simulation sessions conducted in this study.

Nascimiento et al. [41] stated that the affective domain (the third component of the triad for competence training in nursing according to Bloom’s taxonomy [42], which also comprises cognitive and psychomotor domains) was developed during the debriefing phase conducted in clinical simulation sessions. It intensifies nursing students’ motivation to learn, as it involves all the actions determined by Bloom’s taxonomy during the reflective process [43]. Therefore, these authors justified the importance of debriefing for the development of clinical competence in nursing [41]. The positive effects of the debriefing phase on nursing students’ learning process have been widely demonstrated [44]. However, the importance of the expository phase of debriefing included in the MAES© methodology should be emphasized, as a way of empowering nursing students in their learning process. So, consequently, our nursing students may have become more motivated during this study. Based on our previous study, conducted using a qualitative design [4], the motivational elements indicated by nursing students after the MAES© sessions (non-directive/imposing style of the facilitator, adequate structure and planning of the sessions, the possibility of transferring what was learned to the real world, and especially the atmosphere created in the simulation session) were included in the simulated sessions conducted in this study. These motivational elements are intrinsic in the MAES© methodology, and thus the positive results obtained in this study may be a consequence of the characteristics of this methodology.

It is true that significant increases occured in dimensions such as self-efficacy within the motivational components. We believe this is because self-efficacy, as a concept, refers to the perceived ability to perform a task competently and independently. This may be due to the self-directed learning structure itself. It is this aspect that would make the difference for MAES compared to other simulation learning methods. However, our students improved, but not statistically significantly, in the motivation components related to intrinsic value and test anxiety. The intrinsic value of motivation encompasses students’ goals and beliefs for performing the task, i.e., their reasons for performing the task. It is possible that the introduction of a new, unfamiliar educational model is the cause attributable to this aspect (as in other studies of satisfaction with MAES in established groups, intrinsic motivation was a highly valued aspect). Meanwhile, the anxiety component refers to students’ affective and emotional reactions to the task, relating to students’ perceptions of their competence to carry out their work. We believe that the non-significant increase in the score for this dimension could also have the same explanation (the introduction of novelty in the learning routine is often initially accepted with caution by students).

As for critical thinking, our students improved their levels in the four components of this dependent variable according to Alfaro-LeFevre’s model [45]: personal characteristics, intellectual and cognitive abilities, interpersonal abilities and self-management, and technical abilities. Our nursing students achieved significant levels in its four components. Although no study has to date adopted Alfaro-LeFevre’s model [45] to evaluate critical thinking after clinical simulation sessions, different studies have partially examined all of these components, as they are common to the main critical thinking models, such as critical thinking based on Tanner’s clinical judgment model [46] and Lasater’s clinical judgment rubric [47]. In this way, personal characteristics constitute a pattern of intellectual behavior (attitudes, beliefs, and values) that function as an activating element in thinking ability [21]. This component has been identified as confidence [48,49] and self-efficacy [50] by different studies, showing the effectiveness of clinical simulation methodology for improving the self-reported levels of these personal characteristics. As for intellectual and cognitive abilities, this component comprises knowledge of actions and understanding linked to the nursing process and decision-making [21]. Different studies have shown increased levels of knowledge after clinical simulation sessions [51,52,53]. Regarding interpersonal abilities and self-management, these abilities allow therapeutic communication and obtaining information that is relevant to the patient [21]. These abilities have also been demonstrated to be improved after a simulation exercise [54,55]. Finally, the fourth component, technical abilities, are defined as the knowledge and expertise in procedures that are part of the discipline of nursing. Clinical simulation methodology has proven to be useful for improving these abilities and clinical skill performance [50,51,52]. Therefore, the results obtained in our study are congruent with previous studies that also showed significant improvement in critical thinking after high-fidelity simulation [14,15,16,17,26,27,30,31,32,33,38,39,40,41,42,43,44].

It must be highlighted that critical thinking is considered an essential component in nursing practice for acquiring nursing competencies [56] and providing safe and competent care [57]. Furthermore, it is considered an essential competency for newly-licensed nurses who will need to make independent decisions in clinical practice [58,59,60]. Research indicates that nurses with good critical thinking skills can have a positive impact on patient outcomes [61]. Conversely, nurses showing poor critical thinking skills often fail to recognize clinical deterioration, which can result in compromised patient safety [62]. However, the terms critical thinking, clinical reasoning, clinical judgment, problem-solving, and decision-making are often used interchangeably [59,60,63]. This synonymous use of multiple terms to describe how nurses think is one of the primary barriers for the correct identification of effective teaching strategies and associated outcome measures [64]. Additionally, different frameworks have been proposed to define these constructs using various evaluation tools, even within the same frameworks [64].

According to the review conducted by Brown Tyo and McCurry [61], clinical simulation methodology is considered an effective educational strategy that improves critical thinking skills of nursing students, particularly when a structured critical thinking framework or model is adopted, and when the clinical simulation sessions are conducted including a structured debriefing phase [65,66], which incorporates the use of reflection, a proven strategy for promoting higher-level thinking [67,68]. Moreover, clinical simulation sessions that include standardized patients have been demonstrated to be useful for improving critical thinking skills [69]. In this sense, we adopted the well-established 4-Circle Critical Thinking Model of Alfaro-LeFevre [45], standardized patients were included in our clinical simulation sessions, and all the MAES© sessions concluded with a structured debriefing phase, thus fulfilling all the requirements defined by the INACSL Standards of Best Practice: Simulation^SM^ [40] for conducting high-fidelity simulation sessions. It should be noted that Theobald et al. [70] indicated that exposure to a single simulation did not significantly increase critical thinking. However, one study found that three exposures produced a statistically significant increase in critical thinking [48]. Nonetheless, a recent review [71] concluded that the use of rubrics and questionnaires reduced bias and increased objectivity in the simulation setting, and in light of this, we applied a validated questionnaire.

The main limitation of our study is related to the self-reported questionnaires used to evaluate nursing students’ motivation and critical thinking. However, these questionnaires have strong internal validity, and the MAES© sessions were demonstrated to be a useful tool for improving the levels of these dimensions among nursing students. The small sample size was also a limitation in the study. It is possible that the COVID-19 pandemic could have negatively influenced the collection of data.

Finally, the positive effects of the MAES© methodology on developing both motivation and critical thinking in undergraduate nursing students should be confirmed by future research, and therefore further studies are needed. These future studies should broaden the sample and compare it with a control group, using quasi-experimental or experimental designs and evaluating the outcomes obtained in follow-up periods (for instance, after three, six, and/or twelve months). Additionally, future research should also assess the improvement in motivation and critical thinking of nursing students or registered nurses using the MAES© methodology, and extend it to other settings and education centers.

## 5. Conclusions

The MAES© methodology allows undergraduate nursing students to increase their levels of motivation and critical thinking. Specifically, the students improved their levels of motivation components (including self-efficacy, strategy use, self-regulation) and critical thinking components (including personal characteristics, intellectual and cognitive abilities, interpersonal abilities and self-management, and technical abilities). These improvements could be a result of the intrinsic characteristics of the MAES© methodology (such as its team-based, self-directed, collaborative and peer-to-peer learning features). Our results may be confirmed by future research projects using quasi-experimental or experimental designs and follow-up periods, recruiting more nursing students, including registered nurses, and extending the MAES© methodology to other settings and education centers.

## Figures and Tables

**Table 1 healthcare-10-00927-t001:** Overview of demographic variables (*n* = 77).

		*n* (%)	Missing
Questionnaire	CuPCPE	5 (6.49%)	
	MSLQ44	16 (20.78%)	
	Both	56 (72.73%)	
Gender	Women	67 (87.01%)	
	Men	10 (12.99%)	
Previous experience with simulation	No	63 (81.82%)	1
	Yes	13 (16.88%)	
Previous clinical experience	No	58 (75.32%)	1
	Yes	18 (23.38%)	
Previous education	No/None	36 (46.75%)	
	Yes	41 (53.25%)	
		Mean (SD)	
Age (years)		22.07 (2.175)	1

**Table 2 healthcare-10-00927-t002:** Average scores for CuPCPE questionnaire pre- and post-simulation (mean (SD)) and difference between measurements (% of the maximum score) (*n* = 61).

CuPCPE Questionnaire	Mean (SD *)		Difference (%)	*p*-Value **
	Pre	Post		
Total Questionnaire	340.5 (39.15)	357.5 (34.68)	17 (3.9%)	<0.001
Personal dimension	121.9 (13.46)	124.4 (12.67)	2.5 (1.6%)	0.033
Intellectual and cognitive dimension	137.9 (17.46)	146.6 (15.81)	8.7 (4.9%)	<0.001
Interpersonal and self-management dimension	60.8 (15.81)	65.9 (8.47)	5.1 (6.4%)	<0.001
Technical dimension	19.9 (2.75)	20.6 (2.64)	0.7 (2.9%)	0.031

* SD—Standard Deviation ** *p*-value at 0.05 confidence level.

**Table 3 healthcare-10-00927-t003:** Average scores for MSLQ44 questionnaire pre- and post-simulation (mean (SD)) and difference between measurements (% of the maximum score) (*n* = 71).

MSLQ44 Questionnaire	Mean (SD *)		Difference (%)	*p*-Value **
	Pre	Post		
Total Questionnaire	231.9 (20.98)	237.4 (20.84)	5.5 (1.8%)	0.003
Motivation section				
Self-efficacy subscale	237.4 (20.84)	51.9 (6.64)	1.3 (2.1%)	0.039
Intrinsic value subscale	51.9 (6.64)	53.2 (5.57)	0.7 (1.1%)	0.189
Test anxiety subscale	53.2 (5.57)	47.6 (7.38)	0.4 (1.4%)	0.354
Total Subcale	47.6 (7.38)	48.3 (6.69)	1.6 (1%)	0.155
Learning Strategies Section				
Use of learning strategies subscale	48.3 (6.69)	18.9 (5.34)	2.4 (2.6%)	0.001
Self-regulation subscale	18.9 (5.34)	18.5 (5.87)	1.5 (2.4%)	0.030
Total Subcale	18.5 (5.87)	118.4 (12.77)	3.9 (2.5%)	0.002

* SD—Standard Deviation ** *p*-value at 0.05 confidence level.

## Data Availability

The data are available upon email request to the corresponding authors.

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
