# Peer review of "Does Self-Directed Learning with Simulation Improve Critical Thinking and Motivation of Nursing Students? A Pre-Post Intervention Study with the MAES© Methodology"

_healthcare, 2022, doi:10.3390/healthcare10050927_

Round 1

Reviewer 1 Report

The authors have sincerely addressed my concerns. I still have strong reservation on the significance of the work presented. I concur with the recommendation of reviewer#1 to accept this paper as a short communication paper if possible. 

Author Response

Thank you for your time and for reviewing our article. We appreciate your suggestions and comments. We have checked the article again to find errors and try to improve it.

Reviewer 2 Report

The English is very good and the statistics are well done.  I would also say that the particular methodology is particular to part of Spain only and may not be generalizable.  This limits the article's general interest and importance.  In addition, the use of a facilitator and extra time with students will always help with motivation.  Still, once there is a final draft, I can recommend publication, even though interest in the article will be limited.

Author Response

Thank you for your time and for reviewing our article. We appreciate your suggestions and comments. In the limitations section of the study we set out the issues you comment on. We have checked the article again to find errors and try to improve it.

This manuscript is a resubmission of an earlier submission. The following is a list of the peer review reports and author responses from that submission.

Round 1

Reviewer 1 Report

One-armed design without a control group is a fatal limitation of the study. The authors argued that their study may help provide initial idea on the topic and there have been many similar studies, which is not at all persuasive. However, given that the study does provide some preliminary findings that may inform future studies, if the editor agrees, the manuscript may be published as a short communication paper rather than a full article.

Reviewer 2 Report

First of all, I apologize for mis-state the criteria for the null hypothesis rejection. It is indeed the opposite of what I stated. However, Table 3 in the manuscript has 4 rows with p-value < 0.05, and 3 rows with p-value > 0.05. Yet, the manuscript simply made a blanket statement regarding the positive effect of the tool. I expect that the three rows with p-value > 0.05 to be thoroughly analyzed. Also, the entire motivation section total subscale has a p-value of 0.155, which is greater than 0.05. Therefore, the null hypothesis actually holds! Of course, it is not clear how the total subscal calculated and whether or not the total has any meaning at all.

Reviewer 3 Report

All my prior concerns have been addressed.  Well done!